# Design of a Computed Tomography Automation Architecture

**Nicholas Hashem [1],\*, Mitchell Pryor [1], Derek Haas [1] and James Hunter [2]**

1   Department of Mechanical Engineering—Nuclear Engineering, University of Texas at Austin, Austin, TX 78712, USA; mpryor@utexas.edu (M.P.); derekhaas@utexas.edu (D.H.)
2   Los Alamos National Laboratory, Los Alamos, NM 87545, USA; jhunter@lanl.gov
\*   Correspondence: nhashem@utexas.edu

**Abstract:** This paper presents a literature review on techniques related to the computed tomography procedure that incorporate automation elements in their research investigations or industrial applications. Computed tomography (CT) is a non-destructive testing (NDT) technique in that the imaging and inspection are performed without damaging the sample, allowing for additional or repeated analysis if necessary. The reviewed literature is organized based on the steps associated with a general NDT task in order to define an end-to-end computed tomography automation architecture. The process steps include activities prior to image collection, during the scan, and after the data are collected. It further reviews efforts related to repeating this process based on a previous scan result. By analyzing the multiple existing but disparate efforts found in the literature, we present a framework for fully automating NDT procedures and discuss the remaining technical gaps in the developed framework.

**Keywords:** computed tomography; radiography; automation; robotics; radiation imaging; X-ray





## 1. Introduction

Advancements in radiography and computed tomography (CT) techniques are ongoing for applications that span many industries. Recent developments that focus primarily on defect inspection, artifact reduction, or metrology can apply to a breadth of radiography and CT inspection applications. The increased incorporation of automation in various steps of the CT process has been an area of research that opens the door for more effective and efficient non-destructive testing (NDT). This paper reviews recent efforts to automate aspects of radiation imaging of NDT and presents an overall CT architecture based on this review of recent literature.

Industrial radiography is a non-destructive testing method of acquiring 2D images of non-biological samples. Many applications require more information than a single 2D image and therefore use X-ray computed tomography to reconstruct 3D representations of samples by taking hundreds or thousands of single 2D images at different projection angles. Normally, a sample is circularly rotated 360° on a stage while images are captured. This contrasts with medical radiography where the source and detector are mobile while the patient is stationary. Limiting total dose is a key goal and requirement in medical applications, which restricts the possibilities for source types, strengths, and geometries that are otherwise available for industrial applications. Therefore, the focus of this paper is geared toward industrial efforts because of inherent differences between the two areas of application. There are a wide variety of uses for industrial CT, from food inspection [1] to security monitoring for terrorist activities [2]. Certain applications face specific challenges, and various novel CT methods are being implemented to meet these needs.

The general approach to computed tomography (CT) follows a few baseline steps. The components necessary include a sample object that will be imaged, the radiation source, and a detector that collects the radiation after passing through the sample object. Radiation sources used for imaging can vary widely, but the geometric aspect of the beam can be

grouped into parallel, fan, or cone beams (Figure 1). Detectors used for CT are exclusively digital, as analog methods can only be used for 2D imaging applications.

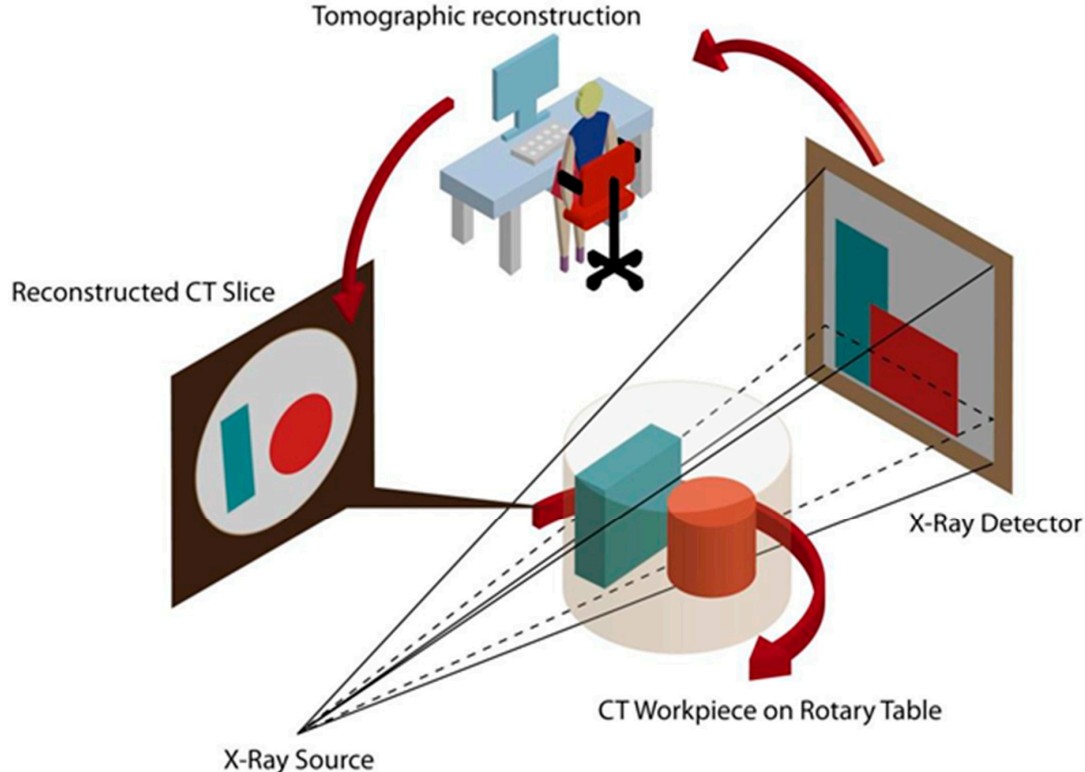

**Figure 1.** Generic X-ray computed tomography (CT) process with cone-beam geometry. Reproduced, with permission, from Reference [3].

An important step in any CT data collection process starts with determining how to orient the sample. It must be between the source and detector, but the distance from each and the angular orientation of the sample are variable. For CT, many images are taken at different angles, and therefore, an axis of rotation must be determined. The holding and transitioning of the sample between each angle are classically accomplished via a rotary stage. Thus, the staging platform limits the choice of axis due to its own kinematic constraints and potential to interfere with the image area.

The imaging procedure is conducted by acquiring projections at small angular steps as the sample is rotated about its chosen axis. The step size and number of projections are chosen depending on the quality of CT data required, the amount of time allowed for the CT process, and the size of the part relative to the size of the detector. A smaller step size through 360° will help to improve 3D CT data quality but will add to the total processing time and size of the imaging data. These 2D projections are then put through a reconstruction algorithm to produce 3D CT data.

This basic summary of the methodology and the simple procedure shown in Figure 2 hides several manual tasks necessary to generate a useful result. These manual tasks include, but are not limited to the following:

- Deciding the position and orientation of the object (and detector);
- Selection of X-ray source energy and beam filtration;
- Precise placement of the object (and detector) in the chosen position and orientation;
- Reorientation and repositioning of the object between different scans;
- Post-processing and analysis (e.g., defect detection).

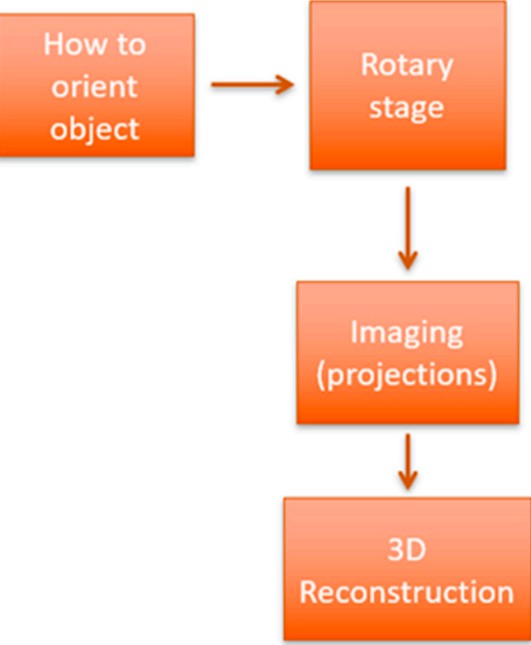

**Figure 2.** Generic computed tomography (CT) procedure that is completed manually.

Manually completing these tasks can adversely affect the quality of the result, are time consuming, prone to error, and may require a repeated startup and shutdown of the radiation source due to the operator entering and exiting the imaging location. For these reasons, various efforts have been undertaken to automate aspects of CT. This review summarizes and synthesizes the published efforts that attempt to minimize these issues through automation. Much of the relevant research in this field focuses on accomplishing one of the following goals: defect inspection, artifact reduction, metrology, calibration, or achieving a faster or improved CT method. A literature review focusing on automation efforts enables us to synthesize the collective efforts into a complete architecture for automated CT. This review of the state-of-the-art research and the use of industrial CT then allows us to identify any remaining gaps in which contributions are needed to create a fully automated CT solution. This review of NDT is primarily focused on recent CT research in order to develop a framework for automated CT and therefore does not attempt to cover the full scope of NDT in industry or research.

First, the research is presented from the review of applicable papers in the literature. Categorization of these papers is primarily based on where in the CT process modifications are automated. These groupings were chosen to help organize the research into the areas where they impact the generic CT procedure. Partial architectures are shown for a selection of papers to serve as an example of how the review of literature informs the creation of a CT framework. Space prevents the inclusion of partial architectures for all applicable cited papers, but the three selected represent the three different types of modifications in the illustrative form of different aspects of the CT process. We then identify gaps and propose a fully automated CT framework.

## 2. Computed Tomography Setup and Initial Methods

This section covers research methods that are implemented prior to acquiring sets of 2D images for computed tomography. This includes research that takes advantage of additional knowledge of the part geometry before imaging, changes to the CT setup, and modifications to the generic scanning procedure.

### 2.1. Prior Restriction of View Angles

Presenti et al. [4] incorporated CAD data to help perform fast in-line defect inspection. They assumed that there is prior knowledge of both the CAD data of the object and the

inspected object itself. This is not always the case because part drawings may not always be available. Presenti et al. built CAD-simulated projections from the 3D CAD model and eliminated much of the time involved in the CT process by restricting imaging to optimal projection angles. This was carried out by defining regions of interest (ROIs) assuming the position and extent of relevant defects are approximately known. Optimal projection angles for seeing defects were found for these ROIs based on image contrast in the CAD model. The orientation and alignment of the sample were then determined from these simulated CAD projection data. Only a few projection angles were selected for which a potential defect is most visible. The projections were then acquired and directly compared with their CAD-simulated counterparts. A chosen threshold for root-mean-squares error difference between the simulated and real projections was used to classify the sample as defective or not. By directly identifying defects in the projection-space, they eliminated the need to perform 3D reconstruction and reduced the time required for CT analysis. Figure 3 is an example of a partial architecture that has been modified by introducing initial data that change the setup and procedure compared to the generic CT procedure in Figure 2 and represent the method used by Presenti et al. [4].

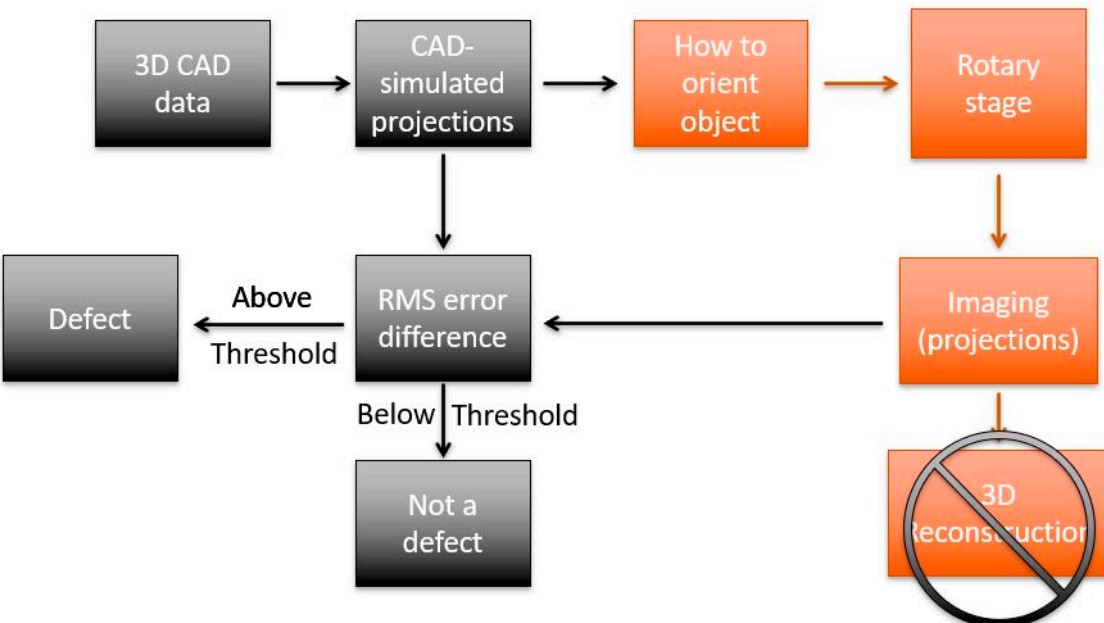

**Figure 3.** Partial architecture based on "CAD-based defect inspection with optimal view angle selection based on polychromatic X-ray projection images" by Presenti et al. [4].

James and Edwards [5] applied robot kinematic modeling for controlling the positioning of parts in beamlines. The focus was to help with parts of complex geometries to be rapidly and accurately oriented in the beam using arbitrary serial robotic manipulators. The aim of their study was to provide a framework allowing for simulation and control of various types of positioning systems, and automatic alignment of parts. Count time minimization was also a focus of this study, which was accomplished by determining the incident beam paths that undergo less attenuation through the material.

### 2.2. CT Setup Positioning

Linear placement of the sample between the source and detector is a focus of multiple projects. Grozmani et al. [6] investigated the influence of sample placement on the uncertainty of measurements in industrial CT. They optimized the object placement using its stereolithography (STL) model to find the placement that minimized the attenuation power. This reduced the presence of overlapping metal regions in the scan. With the STL model and object orientation determined, a minimum source-to-object distance was calculated

that required the object to be fully inside the extent of the cone beam through a complete rotation about the chosen axis. A minimum photon energy was also determined. The minimum source-to-object distance maximized the distance from the object to the detector and resulted in a magnification of the image. The voxel size of the CT scan was decreased, and they claimed that resolution was increased as well. However, the cost of increasing the geometric unsharpness value when this magnification method is used was not mentioned in this paper.

Illemann et al. [7] experimented with varying the sample's linear placement between the source and detector. They focused on the precise source, object, and detector positioning based on the source energy. The pixel magnification was measured while shifting the sample rotary stage forward and backward. The effective position of the source and detector plane varied depending on the source spectrum, the absorbing characteristic of the object, and the detector material and thickness. This was based on the varying penetration depth in the detector, which is where the incident radiation interacts with the scintillator.

Siewerdsen and Jaffray [8] present an additional theoretical method to identify an optimal imaging geometry. They include modifications to the source, object, and image plane distances as well as looking at the X-ray focal spot size in order to optimize the magnification of the imaging.

### 2.3. Non-Circular Scanning Trajectories

Vienne and Costin [9] had the goal of reducing CT acquisition time for direct use in product lines by using a few-views CT inspection. They used robotic systems to allow for the inspection trajectory to not be limited to one acquisition plane, reducing the necessary projections for CT reconstruction. Instead of using the classical circular trajectory for CT, they used a partial spherical trajectory. The use of this complex 3D trajectory required the use of 3D iterative reconstruction algorithms. They focused on using the simultaneous algebraic reconstruction technique (SART) and the discrete algebraic reconstruction technique (DART) (integrates prior knowledge on the object) algorithms. The X-ray generator source and detector were attached to the end of two robots that were controlled via master–slave technology. The acquisition/source points were distributed on a spherical surface equidistant from the sample and on opposite sides of each other to keep a constant magnification. A partial spherical trajectory was defined by fixing a limited angular range for two angular parameters. Their results concluded that an angle out of the circular plane (40° for their experiment) improved the reconstruction of object horizontal edges.

Helical CT scanning is useful when attempting to scan objects that do not fully fit in your imaging area. Brinek et al. [10] used a helical trajectory to perform X-ray CT for correlative imaging. They used helical CT because they had long objects that exceeded the dimensions of the detector. Using helical CT shortened the scanning time over having to take multiple classical circular CT scans and stitching them together. It is important to note that different artifacts are seen in helical trajectories than in circular trajectories. Some artifacts present in a circular CT scan of a sample will not show up at all in its helical scan. Phantoms designed with the focus on the oil and gas industry were used in this study, and it was determined that the image quality of the top sample surface was improved with helical CT. This was because artifacts present in the circular scan performed had a larger presence on or near the top surface of the sample.

Other trajectory optimization methods are presented in several studies that are based on prior object information. Fischer [11] determines the scan trajectory based on the CAD model of the object, optimizing their algorithm based on the same foundations described in Section 3.2 [12].

### 2.4. Source Types

Industrial radiography sources vary in geometry, energy, strength, and radiation type. Discussion of the different beam geometries was discussed earlier—parallel, fan, or cone beam. Energies for X-ray industrial radiography vary widely, usually from ~30 keV to

several MeV [13]. X-ray tubes normally are used for the low and medium source energies up to 450 keV. Radioactive isotopes can also be used in radiography, which require no electrical power supply. Some of the radioisotopes (and average energy levels) used include Tm-170 (72 keV), Yb-169 (200 keV), Ir-192 (450 keV), Se-75 (320 keV), Co-60 (1250 keV), and Cs-137 (660 keV). MeV-level radiography normally requires a linear accelerator to be used as a source. Neutron radiography utilizes a separate radiation source type and is used because materials have different attenuating properties for neutrons and photons. Neutron radiography can be performed in beamline facilities at many nuclear research reactors, spallation source facilities, or independent from large facilities with fusion neutron generators [14].

Beamline facilities at nuclear research reactors can also be used for photon radiography. For example, a MeV-level photon imaging facility was built and demonstrated at the University of Texas at Austin's "Training Research and Isotope Production General Atomics" (TRIGA) MARK-II research reactor [15]. This illustrates the flexibility in source type that can be used for industrial radiography, although the strength and geometric resolution capabilities of the reactor source fall short of those that can be achieved using a linear accelerator. An X-ray and gamma-ray radiography facility was set up and the 7 DOF SIA5D robotic manipulator has since been used to perform automated CT acquisition with a scintillator and digital charge-coupled device (CCD) camera.

## 3. Feedback Techniques

The following papers focus on CT techniques that incorporate initially gathered data into a feedback loop for the purpose of improving imaging parameters and results.

### 3.1. Artifact Reduction

The presence of artifacts in CT data can disrupt inspection processes and reducing them is an ongoing research focus. Artifacts refer to something seen in the image that is not actually there, showing up most notably during 3D visualization. Kano and Koseki [16] focused on reducing artifacts due to the metal pieces in electronic devices. They used a triaxial rotation mechanism to allow for the adjustment of their samples. An initial single-axis rotation scan was performed as a baseline, and the total area of all-metal regions was counted in pixels for each projection. Since decreased metal area means metals are overlapping, one- or two-axis angles were changed so that the metal area increased. This comes from the idea that transmitted X-ray intensities are only saturated in a certain direction because of overlapping metals, and there could be other directions that contain regions that are not saturated. When plotting the area of metal regions vs. projection number, there existed local minima where metals were presumed to be overlapping. Optimum angles of orientation were chosen using the triaxial rotation mechanism so that these metal regions overlapped as little as possible. The maximum rotation used for the non-primary axis was 30° from their initial angle. Additionally, they connected the combinations of the three axes using spline interpolation and completed a control function for the triaxial rotation. A forward projection calculation was conducted based on this control function, and it was expected that the metal area on the transmitted image would decrease. For 3D reconstruction, another algorithm should be used since there are multiple axes used in scanning. They proposed an effective reconstruction algorithm based on the Feldkamp–Davis–Kress (FDK) algorithm modified for multi-axis scanning. Note that these experiments were performed only in simulation. Figure 4 represents the method used by Kano and Koseki [16] in their study and is an example of a partial architecture that has been modified by introducing a feedback loop that serves as an added capability to the generic CT procedure in Figure 2.

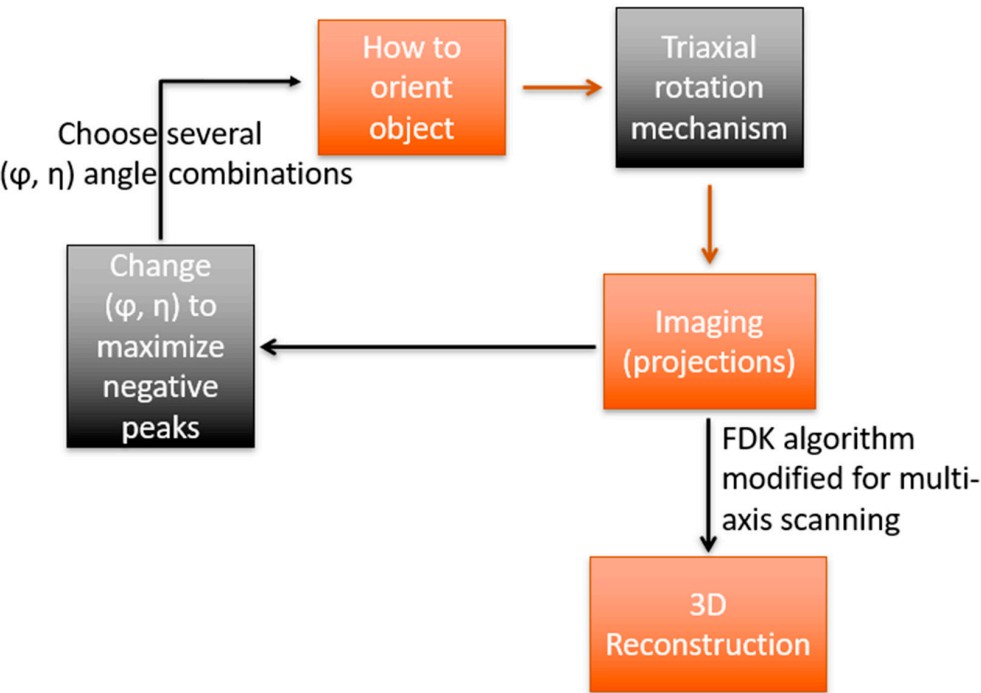

**Figure 4.** Partial architecture based on "Optimization of multi-axis control for metal artifact reduction in X-ray computed tomography" by Kano and Koseki [16]. φ and η are the two rotation angles in voxel space, describing the tilts of 3D space on the stage. These are separate from the classical 360° rotation of the stage itself during an individual scan.

### 3.2. Additional Optimization Examples

Finding the perfect imaging angles for 3D reconstruction is a current research focus for many CT experts. The variation in samples' size, geometry, and material makeup will influence where the optimal orientations are located. The motivation for optimizing trajectories that go beyond the limits of a single-axis circular trajectory can be attributed to the mathematical incompleteness of a single circular scan reconstruction, as described in Bartolac et al. [17]. This is due to the "null cone" generated from the Fourier space sampling for circular cone-beam CT, as shown in that study.

Herl et al. [18] chose to improve the scanning trajectory for robot-based X-ray CT. The flexibility of robotic CT allowed the departure from the classical circular trajectory, optimizing the scanning trajectory using an automated quantitative workflow. The algorithm created and tested is based on a quantitative Tuy criterion [19]. A region of interest is first defined to limit the calculations to the designated volume, and then the optimization iteratively checks the projections as they are created based on these Tuy conditions. Similar methods using include approaches by Maier et al. [20] and Liu et al. [21] to analyze trajectories based on the same Tuy conditions. Hatamikia et al. [22] developed a similar method that is based on capturing an initial CT scan, then analyzing the data to form a new trajectory.

The creation of simulated projections instead of experimentally captured projections is performed and used for many optimization methods, as simulated projections are much more convenient at generating many images for testing than actual imaging. For example, simulated projections are used in the approaches of Heinzl et al. [23] for determining penetration lengths, and Buratti et al. [24] for contrast to noise optimization. Ouadah et al. [12], Brierley et al. [25], and Zhao et al. [26] are further examples of the use of simulated CT projections for optimization of reconstruction trajectories. Reisinger et al. [27] optimized several different parameters, including positioning of the specimen between the source and detector, the orientation, the X-ray tube voltage, and the inclusion of a prefilter thickness to achieve the desired contrast. Ouadah et al. [12] introduced a detectability index

based on the noise-power spectrum and modulation transfer function. Zaech et al. [28] used this index on real projections to determine scanning positions during the CT process.

## 4. Post-Imaging Techniques

Computed tomography results are calculated 3D representations of parts being imaged. There are various post-imaging techniques that can be performed beyond the reconstruction step that help improve the accuracy and efficiency of 3D representations ultimately created.

### 4.1. Artifact Reduction

Herl et al. [29] focused on reducing artifacts due to high absorbing metal parts in X-ray CT scans on multi-material objects with high photon attenuating metal parts. Their solution suggested using projection data from multiple scans with differently positioned object orientations and fusing the data. They used the example of just two different scans with different axes of rotation. They took both scans and independently produced two separate 3D reconstructions. From this, they estimated the local quality of the resulting volumes and then fused these volumes into an optimized volume. This approach was based on the idea of creating more reliable information and ignoring unreliable information. The voxels that were more attenuated in one scan were determined to be the less reliable ones, and hence the other was chosen to optimize the volume. This method did not determine the optimal object positioning, and instead purely relied on increasing the number of CT scanning positions. An additional approach proposed in this research was to introduce shrinking merged algebraic reconstruction technique (smART) and merge sinograms of different scans, estimate the reliability of each projection pixel and then reconstruct the merged sinogram with an iterative reconstruction algorithm. Figure 5 is an example of a partial architecture that has been modified after the imaging step has been performed by changing the way the 3D volume is generated. This can be compared to the generic CT procedure in Figure 2, and it represents the method used by Herl et al. [29].

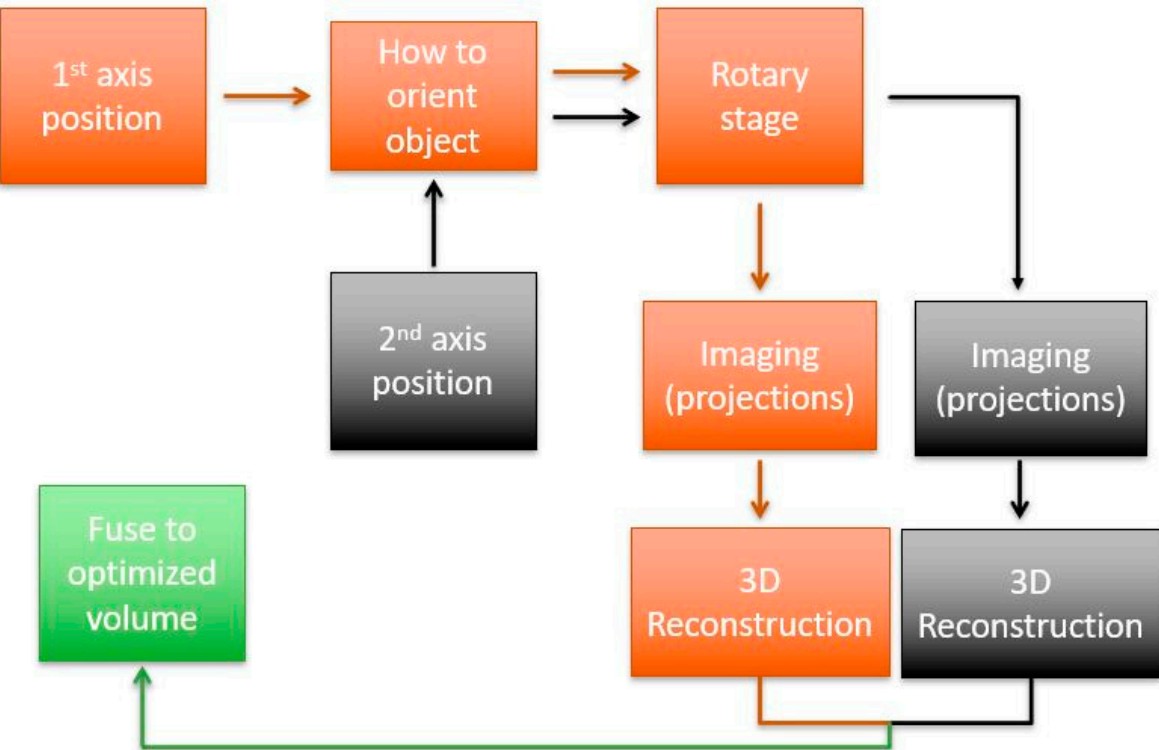

**Figure 5.** Partial architecture based on "Artifact reduction in X-ray computed tomography by multi-positional data fusion using local image quality measures" by Herl et al. [29].

*4.2. Post-Processing Automation*

Automation incorporated into the analysis of CT data was proposed for optimizing and planning in lumber production by Bhandarkar et al. [30]. The analysis of CT images of wooden logs was conducted using machine vision algorithms for identifying and localizing internal defects. Automatic grading of the lumber products using National Hardwood Lumber Association rules was conducted by using the 3D reconstruction with simulated sawing operations. Sawing strategies were then created from this optimization for the highest value yield recovery of the lumber products. There were two separate contributions made. First, a framework for detection, localization, and 3D reconstruction of internal log defects was proposed. Second, models and algorithms were proposed for lumber production optimization that uses the data gathered from the detected defects. The main goal was to automatically produce a decision aid for planning on how to optimally saw the logs.

The food industry also has incorporated automated analysis of CT data. For example, the Danish Meat Research Institute (DMRI) uses CT to measure meat–fat distribution in pig carcasses, and this spatial distribution can deliver an optimal path for a robot to automatically cut the pork for a slaughterhouse. "PigClassWeb" is the system that handles the large amounts of CT scans acquired by DMRI and enables virtual cuts in a reference pig. There is also an accurate estimate of the weights of cuts before they happen on each carcass.

YXLON International's MU56 TB [31] is a robotic industrial X-ray system developed for NDT of turbine blades. This system allows for automatic X-ray inspection without operator interaction. Beyond this, YXLON created "TrueInspect ADR," which is an automatic defect recognition software that can be incorporated with the automatic robotic X-ray inspection. The robot handles, moves, and orients the part throughout the CT scan, while the source and detector remain fixed.

Mery and Filbert [32] demonstrated a method for automating flaw detection based on tracking initially identified potential defects. A sequence of X-ray images was taken at several different orientations and compared with an error-free reference image. Flaws were first detected where there lies a significant difference in pixel regions. The key part of this study was to track these potential flaws through the differently oriented images to rule out false positives if they are not detected from most views. The tracking of flaws was accomplished by identifying the projection of the 3D coordinates of the sample onto the 2D coordinates of the X-ray image's pixels, and the coordinate transformation between the differently oriented images to match one image's pixels to their representative pixels in a separate image. This matching of pixels between separate images is called correspondence, which is solved using epipolar geometry. The geometric transformations and correspondence formulas are described in greater detail in Chapter 3 of Mery's book [33]. Mery and Arteta have also more recently worked on incorporating computer vision techniques into X-ray testing for the goal of automating the detection of defects [34]. That contribution included the creation of a dataset of 47,520 small X-ray images (defects and no-defects), the evaluation of 24 computer vision techniques (deep learning, sparse representations, local descriptors, texture features, etc.), and the MATLAB code used to perform the experiments.

## 5. Industrial Applications

Notable industrial applications of CT can be found in a variety of areas, including the automotive and medical industries. These industries have very different requirements, but robotic automation is adaptable to these challenges.

*5.1. Automotive Industry*

Automated CT inspection is also used in the automobile industry. Krumm et al. [35] worked on rapid robotic X-ray CT inspection in automobile production. A "U"-shaped X-ray CT unit was used to inspect the cars, controlled by an automated rotating industrial robot arm. The robot approached predefined positions on an Audi in this study. This

eliminated the need for sample preparation, which was useful since the sample being imaged was of large mass and size. The sample stayed in place, and the robot moved the CT unit to the proper locations all around the sample.

Holub et al. [36] described the RoboCT unit, which was used in BMW's prototype plant. This system was comprised of four cooperating industrial robots, which held the X-ray imaging components, designed to image fully assembled cars. The four robots were Kuka Quantec extra KR90 R3100 HA on two linear axes. The four robots carried two Comet VarioFocus X-ray sources XRS-225VF, a Varex XRD3025 detector, a Perkin Elmer XRD 1621, and a Gocator optical 3D sensor. The geometry in the prototype plant fit the largest BMW models with a 6-m length. RoboCT assisted with the inspection of joints in the car body, such as rivets, screws, or adhesive bonding. The technology's first implementation in the production environment was introduced in this study.

### 5.2. Medical Industry

The medical industry relies on radiographic and CT imaging methods for viewing the internal features of patients. One limiting factor for medical imaging applications that is of much less concern for industrial radiography applications is the need to limit the dose to the patient because humans and animals are much more sensitive to radiation than industrial parts. An interesting CT application is the need to acquire CT images of horses. A solution is the Equimagine System from Universal Medical Systems [37], which is capable of imaging an entire horse in a single series. The two robotic versions are the Equimagine Helios and Equimagine Zeus. The Helios system utilizes two robotic arms for regular 3D CT imaging, densitometry, tomosynthesis, and panoramic imaging. The Zeus system is a four-arm robotic CT unit, which allows for whole-body scanning in minutes, scanning while the horse is in motion and with reduced radiation dosage.

Kageyama et al. [38] analyzed a multi-axis intraoperative angiography unit for percutaneous pedicle screw placement in the lumbar spine. This is basically radiographic imaging of the blood vessels for a procedure that stabilizes the vertebrae by placing screws in the spine. A total of 17 patients over about two years were included in this study. Pedicle screw insertion can be conducted with a combination of a navigation system with a CT device or the mobile CT device Airo. The multi-axis angiography unit Artis Zeego was used in this study to capture CT-like images. The focus was on minimizing the error and determination of the screw angle. The use of this multi-axis angiography unit over the conventional method increased the screw angle accuracy, reduced the inspection time, and increased the radiation dose.

ORBIT [39] was an open X-ray scanner for image-guided surgery that included a robotic arm holding and controlling the position and orientation of an X-ray source, while the patient and detector remained fixed. The first prototype was deployed at the Charité—Universitätsmedizin Berlin in cooperation with Ziehm Imaging GmbH. The ORBIT project follows a similar idea as with Vienne and Costin [9], in that they are limited in their angular range and use an orbital or spherical trajectory instead of a circular trajectory for CT.

### 5.3. Industrial Robots

Industrial robots are used in many applications due to the improved safety of the operators, increased task speed, productivity, and high repeatability. For CT applications, robots can be used to load and unload parts into a CT scanner, re-orient the part between or during scans, or even can be used directly in robot-based CT systems for heavy objects where they either hold the source, detector, or both during the entirety of a scan. Calibration of these robots is necessary to ensure the geometry of the CT scan is set up well for reconstruction.

#### 5.3.1. Robot Calibration

Amr et al. [40] worked on improving calibration for robot-based CT systems. They presented a method for calibrating by using mainly only the projection data of a special

phantom to minimize reprojection error. The projection data were created by the robot-based system itself. A method was applied to the robot-based CT system to solve for the unknown transformation between the two robots that control the X-ray source and detector, and the transformations for offset and rotation of components relative to the end-effector. The Levenberg–Marquardt algorithm, a damped least-squares method, was then applied to minimize the reprojection error using the manually acquired X-ray projections.

Blumensath et al. [41] also focused on the calibration of robotic manipulator systems. The focus of their study was on applications in cone-beam tomography imaging in which complex scan trajectories are achieved with robots. The robotic system in this case held the sample, while the X-ray source and detector remained fixed. Calibration scans were performed that provided geometric information and to link this to the manipulator's positional encoders' data. Optimization algorithms then estimated source and detector location, detector orientation, and the position and orientation of linear and rotational axes of the robotic manipulator system. They also compared the accuracy of the six-axis robot arm manipulator discussed above with a setup comprising of a rotation stage mounted on the top plate of a high precision hexapod mounted on a linear stage. Further research [42] with the same robotic system was used for computed laminography, which is a type of CT normally used for flat samples.

Landstorfer et al. [43] investigated the positioning accuracy of industrial robots for robotics-based X-ray computed tomography. Only one robotic manipulator was tested, the Kuka KR 15-2 robotic arm. They measured the accuracy of an X-ray CT scan with a robotic manipulator using a laser interferometer. They tracked linear and angular errors in five degrees-of-freedom (DOF). The conclusion of this experiment was that this specific robot had repeatability of $\pm0.04$ mm. This concluded that an X-ray CT scanner based on robots can deliver the reasonable image quality and resolution if the system is calibrated well.

Repeatability experiments were performed with the Motoman SIA5D seven-axis robot [44]. The first experiment involved measuring repeatability with a dial indicator, by having the robot repeat a set of moves before pressing the dial. The second experiment was a test of vibrational impact and trajectory tracking. These were completed to validate the feasibility of this robot for NDT applications at Los Alamos National Laboratory (LANL). The requirements include a maximum of 250 μm variance and vibration under 25 μm. Both measures tested below these requirements, with a repeatability of 18 μm.

### 5.3.2. Robot Loading/Unloading

Automation of loading and unloading of parts is carried out with robots mainly for cabinet CT systems. ZEISS (Figure 6) is one of several companies that have commercialized this setup of both the robot and cabinet CT in combination to automate the loading and unloading of parts. Table 1 lists several popular commercially available industrial X-ray CT systems that incorporate robotic automation.

**Table 1.** Examples of commercially available industrial X-ray CT systems with robotic automation.

| Company | Automated System |
|---|---|
| Carl Zeiss AG [46] | VoluMax in-line CT inspection, Automated loading |
| YXLON International GmbH [31] | Robotic system for industrial X-ray inspection of turbine blades |
| GE Sensing & Inspection Technologies [47] | SpeedScan CT 64, High-speed CT scanner and detector system |
| Nikon Metrology NV [48] | Inline production CT—100% part inspection |
| North Star Imaging Inc. [49] | robotiX, Robotic automated X-ray CT scanning system |
| VisiConsult X-ray Systems & Solutions GmbH [50] | XRHRobotStar, Automated defect recognition capable robot-system |

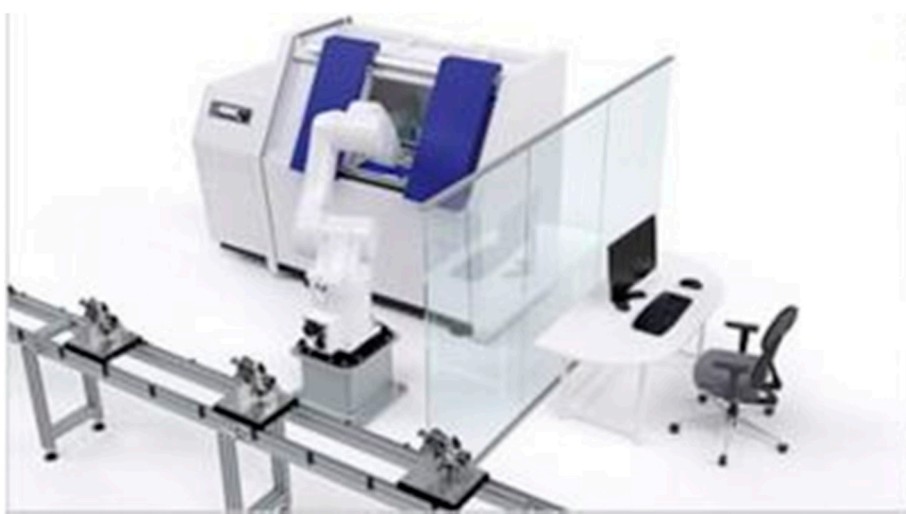

**Figure 6.** Robot and cabinet CT system for automated loading and unloading in a manufacturing line. Ref. [45] Reprinted from CIRP Annals, Volume 63, Issue 2, De Chiffre, Carmignato, Kruth, Schmitt, Weckenmann, Industrial applications of computed tomography, Pages 655–677, Copyright 2021, with permission from Elsevier.

The Advanced Photon Source at Argonne National Laboratory is a high-energy (7 GeV) X-ray synchrotron source that has tested various high-precision positioning techniques [51]. One of these included a robot-based sample-exchange automation system with high positioning repeatability for X-ray cryo-biocrystallography. This robot was used for the changing and realigning of mounted samples.

*5.4. Additive Manufacturing*

As with any method of manufacturing a part, CT also proves to be a useful technique for 3D printed (additively manufactured) parts. As additive manufacturing technologies become increasingly utilized in industry, a majority of the applications can be grouped into either defect detection, dimensional evaluation, density measurement, or roughness analysis [52]. These industrial applications span from musical instruments to historical artifact reconstruction, among others.

Defect detection is a primary application for CT in general and is no different in the case of additively manufactured parts. As new methods for additive manufacturing are becoming increasingly more implemented into the industry, the role of internal inspection by CT becomes important to check the new manufacturing methods for the potential creation of defects such as pores. The use of defect detection for additively manufactured parts with CT is shown in many areas [53–56].

Dimensional evaluation [57,58] and reverse engineering [59] have been performed with the help of CT. These are important in verifying part tolerances and wall thicknesses and can be directly compared with their respective CAD models [60]. Density and roughness measurements and analysis of additively manufactured parts are also performed by CT scans [61–64]. The combination of 3D printed parts with CT imaging also intersects in the medical industry [65]. We can see that many areas are using CT scanning to inspect and analyze their additively manufactured components, but a study on titanium and nickel aerospace components made by additive manufacturing concluded that CT is in fact the best technique for complex geometries [66].

## 6. Computed Tomography Framework

From the review of the literature, recent research has experimented with modifying the baseline steps of the CT procedure, displayed in Figure 2, and with the addition of new components into the process, through methods such as inputting prior information, feedback mechanisms, or post-imaging techniques. Recent research efforts represent

different ways of performing these baseline steps of the CT procedure, and this variability is necessary to apply to as many CT applications as possible. The diagram in Figure 7 is a general CT framework that includes the orientation, imaging, and reconstruction steps of the CT process. Additional blocks represent examples of additional methods that can be applied to the generic procedure based on the literature discussed in this paper.

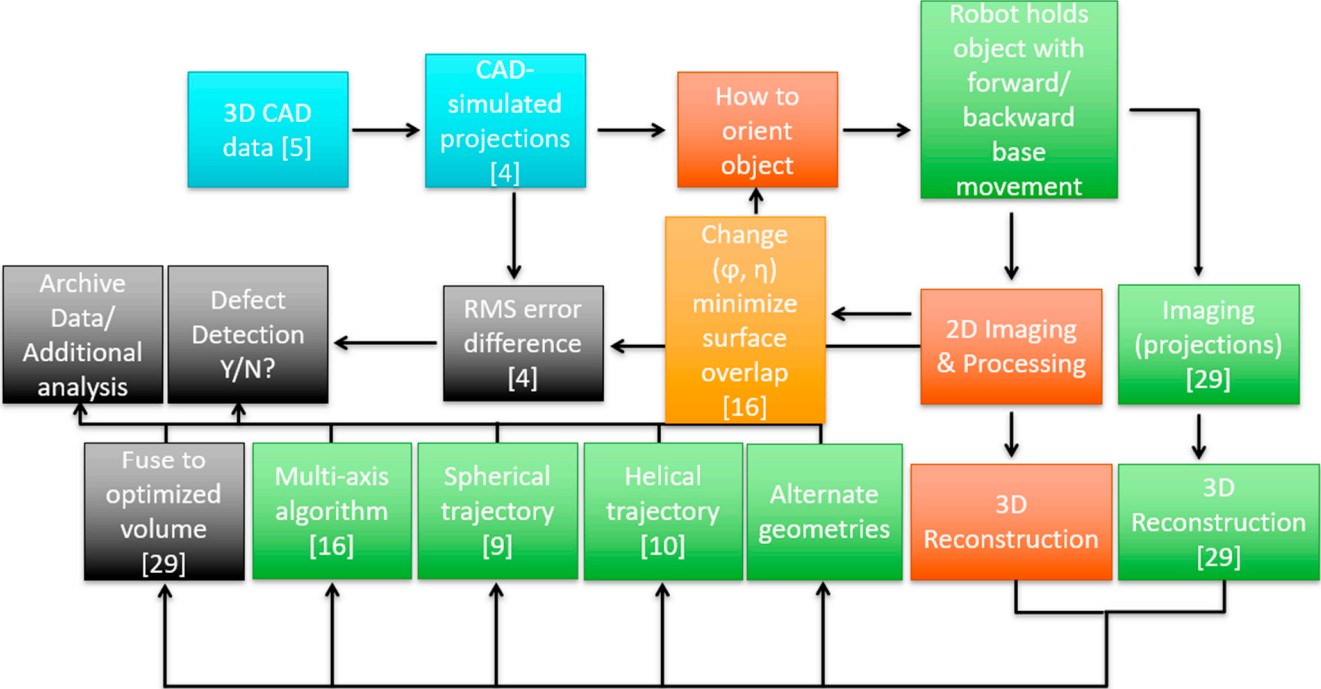

**Figure 7.** Computed tomography framework with the generic CT approach (dark orange), along with additional extensions for certain applications that can be applied. Box colors: prior inputted data (blue), feedback loop (light orange), generic CT approach modifications (green), and post-processing analysis (black). Reference numbers are included as an example for the papers that focus on each aspect or method.

The image acquisition step of the CT process has several parameters that can be modified. Step size, number of projections, exposure time, source type, detector type, and the geometry of the source, object, and detector are some of the most important parameters. The framework that is discussed will identify and implement as many of the necessary image acquisition parameters as possible into its design.

The 3D reconstruction step of the CT process changes based on the selection of an algorithm. There are a few reconstruction algorithms that are seen as most popular in the literature, but the framework will provide an opening for additional algorithms that are not included either due to the lack of documented applicability to current methods or because the algorithms have not been developed yet. Key cone-beam CT reconstruction algorithms are listed in Table 2. Feldkamp's convolution backprojection is the most popular CT reconstruction method for cone-beam flat-panel detectors.

In addition, some techniques in the literature go beyond modifying the way orientation, imaging, and reconstruction are performed. Examples of this include implementing existing information into the orientation decision process prior to any X-ray images being acquired, different physical orientation techniques with additional DOF, the use of more than one CT axis or non-circular trajectories, and adding feedback into the loop for deciding how to reorient the object.

The creation of a flexible framework for industrial X-ray CT applications would help bring together contributions identified individually in the literature. The required components for performing CT, image acquisition, and 3D reconstruction, act as the core for this framework. These base components will offer customizable parameter decisions to

apply to a broad set of applications. Additional techniques will act as add-ons to the base of this framework because its structure will define openings for add-ons to be implemented. This software framework integrates the individual components identified in the literature into a single framework compiled from the reviewed efforts.

**Table 2.** Computed tomography reconstruction algorithms [3].

| Researcher | Exact/Non-Exact | Implemented | Computing Efficiency | Geometry of X-ray Vertices | Reconstruction Method |
|---|---|---|---|---|---|
| Altschuler [67] | N | No | Fair | Circle | Matrix inversion |
| Feldkamp [68] | N | Yes | Excellent | Circle | Convolution backprojection |
| Finch [69] | E | No | n/a | Sufficiently large circle | A mathematical analysis |
| Grangeat [70,71] | N | Yes | Good | Circle | Convolution backprojection |
| Hamaker [72] | N | No | Fair | Finite number of sources | Convolution backprojection |
| Herman [73] | N | Yes | Excellent | Circle | Convolution backprojection |
| Imiya [74] | N | Yes | Poor | Sphere | Rho-filter convolution backprojection |
| Katsevich [75] | E | Yes | n/a | Helical reconstruction | Convolution backprojection |
| Kowalski [76] | N | Yes | Fair | Two parallel circles | Matrix inversion |
| Kowalski [77] | N | Yes | Fair | Straight line | Matrix inversion |
| Minerbo [78] | N | No | Fair | Circle | Convolution backprojection |
| Peyrin [79] | N | No | Poor | Sphere | Rho-filter convolution backprojection |
| Schlindwein [80] | N | Yes | Fair | Two parallel circles | Algebraic reconstruction technique |
| Smith [81] | E | No | Poor | Sphere | Convolution backprojection |
| Tuy [19] | E | No | Poor | Two perpendicular circles | Convolution backprojection |

The core framework can be completed along with a new component addressing a key identified gap. This core framework with the newly developed capability can serve as an example for further opportunities to develop various application-specific solutions more easily. The framework also allows for other gaps identified in the literature to be addressed.

This technique will be implemented in the software framework outlined above. Not all the capabilities discussed will be implemented, but the goal will be to use the knowledge garnered from the review above to create a modular and extensible framework usable by both future researchers and end-users alike to simplify the integration of necessary capabilities.

## 7. Conclusions and Future Work

From this review of literature, there are some common gaps that were not addressed and areas with room for improvement. One area is how to determine the optimal orienta-

tion of a sample for radiography or CT. Radiography would focus on the ideal view angle for 2D images, while CT orientation corresponds to the ideal axis or axes of rotation. The identification of defects is one of CT's most used abilities, especially for internal inspection. Experimental tests for simulation-only projects are lacking in some studies. There is room for increased automation and application-specific solutions. These solutions would benefit from multiple efforts identified in the literature. A framework would allow greater adoption and/or integration of the many techniques described in this paper.

For a feedback loop component, adjustments to CT scan parameters can be based on certain properties of the initial 3D data. The initial data's overall 3D resolution and the abundance of artifacts will determine how small the CT step size needs to be for the required spatial resolution. This will then directly determine the number of projections in the 360° scan. The saturation of the image data will determine any adjustments to the individual exposure times. The signal-to-noise ratio (SNR) can inform the amount of image averaging necessary at each step. Beyond these parameters for a standard CT scan, the reorientation of the part outside of the initially chosen scan axis would be determined based on the defect analysis of the first scan. A robotic manipulator helps to automate and optimize this process. No manual adjustment of the part is required between the initial scan and the subsequent sets of radiographs.

The population of the framework partially includes the use of existing commercial software solutions. Reconstruction, visualization, and analysis can be performed using software currently used at LANL. The LANL-owned software "Recon" is an example of a reconstruction software that can be used to populate that part of the framework. The reconstruction step takes the raw 2D image data from the CT scan and converts it into data that can then be visualized in 3D. After the 3D reconstruction step, this volumetric visualization of the data can be performed with commercial software such as VGStudio MAX [82] or Dragonfly [83]. The Porosity/Inclusion Analysis Module from VGStudio MAX includes a defect detection algorithm, VGDefX, that can locate pores, holes, and inclusions within parts and provides information about these types of defects. Such information includes defect position, size, and geometry. Dragonfly also provides defect analysis in its quantification and analysis tool, including porosity and void analysis. The output of defect detection and analysis performed in existing commercial software can be useful in that it can provide the input to a feedback loop for finding the optimal imaging orientation of the part based on the user's defined metrics.

The creation of the framework will be initially generated by using the generic CT procedures useful for all applications. This core framework, as described previously, will offer flexibility in the parameters that are required to be inputted for the CT process to work, in addition to allowing the option for augmentation of the framework through add-ons. If an end-user has a preferred method for determining the optimal orientation, then the option to modify the prior setup parameters or feedback loop is available. For example, they could use an application-specific algorithm on the images to guide their orientation decisions or use prior CAD data and choose to eliminate the use of feedback.

The software framework will be created within a language or platform. For example, this can be created for the ABB Ltd. robot platform or an open-source language. The imaging and positioning of the sample will be conducted using existing X-ray equipment and robots at LANL. A reconstruction algorithm will also be chosen either from the existing implemented software at LANL or from the existing literature. This choice will be given as a flexible parameter in the software framework because there are many different useful algorithms available.

Once the core of the framework is completed, an example technique will be implemented as an add-on component, acting as a guide for the further population of this framework. This selected area of research will be experimentally verified using LANL CT facilities and will aim to improve upon applications performed at LANL.

**Author Contributions:** Conceptualization, N.H., M.P., D.H. and J.H.; methodology, N.H., M.P. and D.H.; writing—original draft preparation, N.H.; writing—review and editing, N.H., M.P., D.H. and J.H. All authors have read and agreed to the published version of the manuscript.

**Funding:** This research received no external funding.

**Institutional Review Board Statement:** Not applicable.

**Informed Consent Statement:** Not applicable.

**Data Availability Statement:** Data sharing not applicable.

**Acknowledgments:** The authors would like to thank the colleagues at Los Alamos National Laboratory for supporting this research.

**Conflicts of Interest:** The authors declare no conflict of interest.

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
