# Peer review of "Design of a Computed Tomography Automation Architecture"

_applsci, doi:10.3390/app11062858_

Round 1

Reviewer 1 Report

There are some weaknesses through the manuscript which need improvement. Therefore, the submitted manuscript cannot be accepted for publication in this form, but it has a chance of acceptance after a major revision. Number of references (reviewed papers) is not enough for a review paper. My comments and suggestions are as follows:

1- Abstract gives information on the main feature of the performed study, but some details about the main concept of NDT by computed tomography must be added.

2- Authors must clarify necessity of the performed review. Aims and objectives of this review, must be clearly mentioned in the last part of introduction. The “comprehensive review” must be removed from the title.

3- The literature study must be enriched. In this respect, authors must read and refer to the following papers: (a) https://doi.org/10.1007/s10921-020-00721-1 (b) https://doi.org/10.1016/j.powtec.2021.02.072  The current version has only 53 references and it is not enough.

4- Since this manuscript is a review paper, authors must add several figures from reviewed research works. Drawing new figures is not required and adding published figures from previous studies is s necessity.

5- The main reference of each formula must be cited. Moreover, each parameters in equations must be introduced. Please double check this issue.

6- For and industrial applications there are lots of references which must be reviewed and added.

7- In its language layer, the manuscript should be considered for English language editing. There are sentences which have to be rewritten.

8- The conclusion must be more than just a summary of the manuscript. List of references must be updated based on the proposed papers. Please provide all changes by red color in the revised version.

Reviewer 2 Report

A useful and well written top level review of components in a CT system. The future outcomes describing the resulting framework is likely to be of great interest to a lot of CT groups.

A few minor alterations are suggested:

Page 3, line 84 – One would hope that any x-ray CT system is sufficiently shielded, and with interlocks to prevent any radiation exposure. Therefore I am not sure radiation exposure can be claimed as an operator hazard.

Page 6, line 185 – I believe you meant ‘concluded’ rather than ‘included’.

Page 6, line 197 – Did the Brinek et al say why the top sample surface image quality was better with a helical scan? Might be worth mentioning that here.

Page 6, line 216 – I believe you meant ‘spallation’ rather than ‘fusion’ when talking about neutron beam generation facilities.

Page 7, line 245 – You might want to re-phrase ‘these minima were maximised’ to aid clarity. Presumably the ray transmission minima caused by overlaps of highly opaque parts of the sample are undesirable.

Page 8, line 299 – Please clarify why the volume data used ‘cannot’ be raw 3-D images.

Figure 7 caption – The color coding needs to be explained in greater detail.

Round 2

Reviewer 1 Report

The paper has been improved and the current version can be considered for publication.